# Feature-Based Molecular Network-Assisted Cannabinoid and Flavonoid Profiling of *Cannabis sativa* Leaves and Their Antioxidant Properties

**DOI:** 10.3390/antiox13060749

**Published:** 2024-06-20

**Authors:** Ling Chen, Hong-Ling Li, Hong-Juan Zhou, Guan-Zhong Zhang, Ying Zhang, You-Mei Wang, Meng-Yuan Wang, Hua Yang, Wen Gao

**Affiliations:** 1State Key Laboratory of Natural Medicines, School of Traditional Chinese Pharmacy, China Pharmaceutical University, Nanjing 211198, China; cl9426464@163.com (L.C.); honglingli111@163.com (H.-L.L.); 17775481058@163.com (H.-J.Z.); ahyzgz@163.com (G.-Z.Z.); wangmengyuan0202@163.com (M.-Y.W.); 2China National Narcotics Control Commission—China Pharmaceutical University Joint Laboratory on Key Technologies of Narcotics Control, Nanjing 210009, China; 3Institute of Forensic Science, Ministry of Public Security, Beijing 100038, China; ying_zh@126.com; 4Key Laboratory of Drug Monitoring and Control, Drug Intelligence and Forensic Center, Ministry of Public Security, Beijing 100193, China; youmei_626@163.com

**Keywords:** flavonoids, cannabinoids, quantification, antioxidation

## Abstract

*Cannabis sativa* (*C. sativa*) leaves are rich in cannabinoids and flavonoids, which play important antioxidant roles. Since the environmental factors may influence the accumulation of antioxidants in herbal medicines, which affects their activity, this study aimed to investigate the correlation between the chemical composition of *C. sativa* leaves and their geographical origin and antioxidant activity. Firstly, a high-resolution mass spectrometry method assisted by semi-quantitative feature-based molecular networking (SQFBMN) was established for the characterization and quantitative analysis of *C. sativa* leaves from various regions. Subsequently, antioxidant activity analysis was conducted on 73 batches of *C. sativa* leaves, and a partial least squares regression (PLS) model was employed to assess the correlation between the content of cannabinoids and flavonoids in the leaves and their antioxidant activity. A total of 16 cannabinoids and 57 flavonoids were annotated from *C. sativa*, showing a significant regular geographical distribution. The content of flavonoid-*C* glycosides in Sichuan leaves is relatively high, and their antioxidant activity is also correspondingly high. However, the leaves in Shaanxi and Xinjiang were primarily composed of flavonoid-*O* glycosides, and exhibited slightly lower antioxidant activity. A significant positive correlation (*p* < 0.001) was found between the total flavonoids and cannabinoids and the antioxidant activity of the leaves, and two flavonoids and one cannabinoid were identified as significant contributors.

## 1. Introduction

*Cannabis sativa* L. (*C. sativa*), an annual dioecious herb of the Cannabaceae family, is one of the oldest plant sources of food and textile fiber; it was gradually introduced into Europe and became popular for its medicinal use. *C. sativa* was used for the treatment of various intractable diseases but was later banned due to its unstable therapeutic effect and strong side effects [1]. The double-edged-sword-like pharmacological effects have been confirmed from the phytocannabinoids, which were enriched in the female inflorescence. With the discovery of medicinal uses of cannabidiol (CBD) [2,3] and the development of related drugs in recent years, some countries and regions such as the United States and Canada have implemented Cannabis species legalization policies.

*C. sativa* mainly contains two types of components: the cannabinoid family and the non-cannabinoid family [4,5]. Cannabinoids, including tetrahydrocannabinol types (THCs), cannabidiol types (CBDs), cannabigerol types (CBGs), cannabichromene types (CBCs), etc., have exhibited various and noteworthy pharmacological activities in the pharmaceutical field. These activities encompass but are not constrained to anti-epileptic [6,7], anti-oxidation, and anti-tumor [8,9]; thereby offering new possibilities for treating conditions like epilepsy and tumors. Non-cannabinoids such as flavonoids and terpenoids [10,11] play vital roles in plant species [12] and have beneficial effects on human health through actions such as antioxidation and anti-inflammation. Research has shown that the relative contents of flavonoids were variation in *C. sativa* leaves from different regions [13]. Thus, to clarify the correlation between the antioxidant activity and the intrinsic chemical composition of *C. sativa* leaves, samples across different regions were collected, and their chemical and antioxidant properties were profiled. This analysis aims to pinpoint the key components contributing to the antioxidant properties of *C. sativa* leaves.

Feature-Based Molecular Networking (FBMN) [14,15] is an algorithm in the Global Natural Products Social Molecular Networking (GNPS) platform that annotates and identifies compounds by correlating secondary fragmentation information from their mass spectra. On one hand, the FBMN method can cluster compounds based on fragment similarity, and start from known compound nodes for rapid annotation of unknown chemical constituents [16,17]. On the other hand—according to the ionic strength of the compound—semi-quantitative research on the chemical composition can be carried out, and the content difference results can be displayed in the network node [18,19]. FBMN has developed rapidly and is widely used to assist in the structural identification of natural metabolites [20], targeting the discovery of new compounds [18] and revealing the biotransformation and metabolic pathways of the compounds [21].

In this research, we first established an untargeted metabolite profiling method combined with a semi-quantitative FBMN (SQFBMN) approach for chemical annotation in *C. sativa* leaves from five different regions. The findings revealed significant chemical diversity in *C. sativa* leaves among various geographical locations, particularly in terms of flavonoids. To explore the relationship between chemical composition and antioxidant activity properties, 11 flavonoids and two cannabinoids (Δ^9^-THC and CBD) were quantified in the extended *C. sativa* leaf samples. Subsequently, the antioxidant capacity of these *C. sativa* samples was assessed using the DPPH (1,1-diphenyl-2-picryl-hydrazyl radical), ABTS (2,2′-Azinobis-(3-ethylbenzthiazoline-6-sulphonate)), and FRAP (ferric reducing antioxidant power) assays. Moreover, the correlation between the specific flavonoids and the antioxidant activity demonstrated by *C. sativa* leaves was investigated.

## 2. Materials and Methods

### 2.1. Reagents and Materials

HPLC-grade acetonitrile and methanol were purchased from Merck (Darmstadt, Germany) while HPLC-grade formic acid was purchased from ROE (Newark, New Castle, DE, USA). Deionized water was prepared by the Milli-Q water purification system (Millipore, Bedford, MA, USA).

The cannabinoids standards of Cannabidiol (CBD), Cannabidivarin (CBDV), Cannabidiolic acid (CBDA), Δ^9^-Tetrahydrocannabivarin (Δ^9^-THCV), Δ^9^-Tetrahydrocannabinolic acid A (Δ^9^-THCA A), Cannabigerol (CBG), Cannabigerolic acid (CBGA), Cannabichromene (CBC) and Cannabichromenic acid (CBCA) were purchased from Cerilliant (Darmstadt, Germany). The flavonoid standards including Luteolin-7-*O*-glucoside, Luteolin-7-*O*-glucuronide, Apigenin-7-*O*-glucuronide, Vitexin, Vitexin-4″-*O*-glucoside and Vitexin-2″-*O*-rhamnoside were purchased from Purify technology (Chengdu, China).

*C. sativa* samples were provided by local police from eight locations in seven provinces of China; including Baoxing (Sichuan, SC-B), Zhaojue (Sichuan, SC-Z), Hulunbuir (Inner Mongolia, IM), Xi’an (Shaanxi, SAX), Kashgar (Xinjiang, XJ), Yunnan (YN), Xining (Qinghai, QH) and Nanning (Guangxi, GX) (Appendix A). All of the samples were collected during July and October before flowering. The subtypes of the samples were classified based on the active THCA and CBDAS genes using multiplex PCR methods [22]; this showed that the active THCA was highly expressed in *C. sativa* mainly from XJ and IM, while the active CBDAS was highly expressed in others (Appendix A). Plants were air-dried in light-resistant containers at room temperature. The dried samples were crushed and passed through a 40-mesh sieve. The powder was stored at 4 °C before analysis.

### 2.2. Preparation of Standard and Sample Solutions

#### 2.2.1. Sample Preparation for Metabolome Profiling and FBMN

Before preparing the analytical samples, different batches of *C. sativa* from the same region were mixed in equal quantities as the region’s representative sample for method optimization (Appendix A). The leaf powder of *C. sativa* (10 mg) was ultrasonically extracted with 3 mL 75% methanol at 100 kHz for 30 min. After centrifuging at 13,000 rpm/min for 10 min, the supernatant was obtained as the test solution for metabolome and FBMN.

The reference solution was prepared by dissolving the reference standard using 75% methanol into a concentration of about 20 μg/mL.

#### 2.2.2. Sample Preparation for Quantification and Antioxidant Activity Assay

To guarantee the complete extraction of target chemical constituents, the extraction solvents (Appendix A), solid–liquid ratio (Appendix A), extraction time (Appendix A) and extraction frequency (Appendix A) were optimized (using the L28 leaves). The extraction solvents were a mixture of methanol and water in different proportions. The ratio of material to liquid was 10:2, 10:3 and 10:4. The extraction time was 30 min, 40 min, 50 min and 60 min. The extraction frequency was once and twice. The results showed that the optimum extraction condition was as follows: When 3 mg samples were ultrasonically extracted with 10 mL 75% methanol for 40 min and the extraction frequency was once, the total extraction completeness rate reached 83%. 

We accurately weighed 10 mg of fine powder of each sample. The powder was subsequently extracted ultrasonically (40 kHz, 500 W) with 3 mL methanol-water (75:25, *v*/*v*) for 40 min, then centrifuged at 13,000 rpm for 10 min. The supernatant was diluted into an appropriate concentration after adding the IS solution (the final concentration was 10 μg/mL, for quantified analysis) or not (for in vitro antioxidant activity assay). The solution was stored at −20 °C, and centrifuged at 13,000 rpm for 10 min before analysis.

The reference standard solutions for quantified analysis were prepared as below: the reference standards of 13 components and IS were first dissolved with methanol to make the stock solution at 1 mg/mL. Then, the stock solutions were mixed and diluted into a series of concentrations of working solution, in which the concentration of IS was kept consistent at 10 ng/mL.

### 2.3. UHPLC-QTOF MS Analysis Conditions

All samples were performed on a UHPLC-QTOF-MS system consisting of an Agilent 1290 infinity UHPLC system equipped with a binary pump, an online degasser, an autosampler and a thermostatically controlled column compartment coupled to a 6530 QTOF system with a Dual AJS ESI source operating in dual ionization modes (Agilent Technologies, Santa Clara, CA, USA). Liquid chromatographic separation of the *C. sativa* samples was achieved on an Agilent Zorbax Eclipse Plus C18 column (2.1 × 100 mm, 1.8 μm). In the UHPLC system, the mobile phase consisted of 0.1% formic acid in water (A) and acetonitrile (B). The solvent gradient program was set as follows: 0–5 min, 5% B; 5–35 min, 5–20% B; 35–59 min, 20–50% B; 59–79 min, 50–80% B; 79–84 min, 80–95% B; 84–90 min, 95% B; post time, 6 min. The column oven temperature was set at 30 °C, and the flow rate was maintained at 0.3 mL/min. The injection volume was 3 μL in *C. sativa* leaves samples.

The parameters of the MS analysis were as follows: capillary voltage, 4000 V; nozzle voltage, 1000 V; nebulizer gas, 35 psig; fragmentor voltage, 120 V; sheath gas temperature, 350 °C; sheath gas flow rate, 11 L/min; drying gas (N_2_) temperature, 350 °C; drying gas flow rate, 10 L/min. The MS data were acquired in the range of 100–1500 *m*/*z*, and the top 4 ions underwent auto MS/MS in the range of 50–1000 *m*/*z*. Active exclusion was enabled with the following parameters: excluded after 2 spectra; released after 0.5 min. The collision energy (CE) of MS/MS was set at 10, 20 and 40 V.

### 2.4. Data Processing and Construction of Feature-Based Molecular Networks

The raw MS/MS data files were imported into Progenesis QI (Waters, Milford, MA, USA). The adduct ions were selected [M + H]^+^, [M + H-H_2_O]^+^, [M + Na]^+^, [M + NH_4_]^+^ in positive ion mode and [M-H]^−^, [M-H + HCOOH]^−^, [M-H-H_2_O]^−^ in negative ion mode. Filter strength was set to 0.5 and the Peak picking limit was 5, then the results were exported as MSP and CSV files. The MSP files contain MS/MS spectral summary, and the CSV files contain m/z value, retention time, abundance, etc. Finally, the files were uploaded to the Global Natural Products Social Molecular Networking platform for feature-based molecular networking analysis. The parameters were set as follows: Min Pairs Cos: 0.7; Minimum Matched Fragment Ions: 5; Predictor ion mass tolerance: 0.02 Da; Fragment Ion Mass Tolerance: 0.02 Da. Visualization of results via Cytoscape 3.8.1 software. The original full molecular networks in positive mode and negative mode were linked to https://gnps.ucsd.edu/ProteoSAFe/status.jsp?task=5c7ac024e2df47fb8773b095f06673ec (accessed on 9 April 2022) and https://gnps.ucsd.edu/ProteoSAFe/status.jsp?task=8f38924c639c4768bbe65d994c4c138c (accessed on 9 April 2022) separately.

### 2.5. UHPLC-QQQ MS Analysis Conditions

A Shimadzu UHPLC system (Shimadzu Corp., Kyoto, Japan), consisting of a DGU-20A5R degasser, LC-30AD binary pump, a SIL-30AC auto-sampler, a CTO-20AC column oven, and an SPD-M20A UV/VIS detector, was used for the chromatographic analysis. Chromatographic separation was carried out on an Agilent ZORBAX Eclipse Plus C18 analytical column (2.1 × 100 mm, 1.8 μm) within a Security-Guard C18 column (2.1 × 5 mm, 1.8 μm) at 30 °C. The mobile phase comprised 0.1% formic acid aqueous solution (A) and acetonitrile (B) worked as follows: 0–1 min, 12% B; 1–2 min, 12–18% B; 2–9 min, 18% B; 9–13 min, 18–80% B; 13–17 min, 80% B; 17–18 min, 80–100% B; 18–20 min, 100% B; post time was 4 min. The flow rate was 0.4 mL/min, and the injection volume was 1 μL. 

A Shimadzu 8050 triple quadrupole mass spectrometer (Shimadzu Corp., Kyoto, Japan) equipped with an electrospray ionization (ESI) source was used for mass data acquisition. Multiple reaction monitoring (MRM) was performed both in positive and negative ion modes. The optimized MS conditions were set as follows: heating gas flow, 10 L/min; nebulizer gas flow, 3 L/min; desolvation temperature, 250 °C; interface temperature, 300 °C; heat block temperature, 400 °C; drying gas flow, 10 L/min; interface voltage, +4 KV/−3 KV, and detector voltage, −1.98 KV. The optimized MS/MS detection parameters were listed in Appendix A The Shimadzu LabSolutions LC–MS V5.65 software (Kyoto, Japan) was used for data acquisition and analysis.

### 2.6. Method Validation of Muti-Component Quantification

The proposed method was verified following the validation of analytical methods recorded in Chinese Pharmacopoeia (2020 edition) [23].

#### 2.6.1. Linearity and Sensitivity

Calibration curves of each analyte were constructed using its peak area ratios (y) (analyte to IS) versus the analyte concentrations (x) based on weighted least square linear regression (1/x^2^) in the form of y = ax + b. LOD and LOQ are expressed as LOD = 3.3 S/N and LOQ = 10 S/N. 

#### 2.6.2. Precision

Six replicates of QC samples at three concentration levels were determined on 1 day and on 3 consecutive days to evaluate the intra-day and inter-day precision of the method, respectively. The data RSD < 15% met the requirement for analysis.

#### 2.6.3. Stability

In order to determine the accuracy and repeatability, the selected L34 samples were analyzed six times on the same day, and the stability tests were carried out in 0, 2, 4, 8, 16 and 24 h. The stability of the analyte was acceptable when its accuracy bias was 15%.

#### 2.6.4. Accuracy

The accuracy investigation adopts the method of dosing recovery rate: a certain amount of sample powder (L34) is weighed in parallel, 50% of the target component content of the control solution is added, the sample injection analysis of the test solution is prepared, the peak area of each component to be measured is recorded, and the recovery rate is calculated. The results are shown in Appendix A, and the results show that the recovery rate of 13 analytes is between 80 and 105%, indicating that the accuracy of the proposed method was acceptable.

### 2.7. Antioxidant Activity Assay

The in vitro antioxidant activities of the *C. sativa* leaves, which were evaluated against free radicals and oxidants, were quantified by three well-established methods including DPPH (1,1-diphenyl-2-picryl-hydrazyl radical) assay, the ABTS (2,2′-Azinobis-(3-ethylbenzthiazoline-6-sulphonate)) assay and the FRAP (ferric reducing antioxidant power) assays.

#### 2.7.1. DPPH Assay

The ability of *C. sativa* leaves to scavenge radicals was determined using DPPH, according to the procedure described in the literature [24,25]. Briefly, an aliquot of 20 μL extract was mixed with 180 μL of newly prepared 0.2 mmol DPPH solution dissolved in methanol; and a series of Trolox solutions (0.015, 0.03, 0.15, 0.30, 0.60, 0.90, 1.50 mmol/L) was used to calculate the standard curve. The mixture was incubated for 30 min at room temperature, then its absorbance was measured at 517 nm.

#### 2.7.2. ABTS Assay

The ABTS+ stock solution was newly prepared in line with the instruction manual of the commercial kit, and then incubated at room temperature for 16 h in the dark and used within 2 days. The stock solution was diluted using 75% ethanol to prepare the working solution with an absorbance of 0.70 ± 0.05 at 734 nm by a microplate reader. An aliquot of 5 μL of the extract or Trolox solution (0.015, 0.03, 0.15, 0.30, 0.60, 0.90, 1.50 mmol/L) was mixed with 200 μL ABTS+ working solution. After 5 min incubation in the dark at room temperature, the absorbance of the mixture was measured at 734 nm. 

#### 2.7.3. FRAP Assay

An aliquot of 5 μL of extract solution was mixed with 180 μL of newly prepared FRAP reagent. After incubation at 37 °C for 5 min, the absorbance of each reaction mixture was monitored at 593 nm. FeSO_4_ was used as the positive control. The antioxidant activity corresponded to the increased absorbance during reducing Fe^3+^ into Fe^2+^, and the result was expressed as the concentration of antioxidants having a ferric-reducing ability equivalent to that of 1.0 mmol/L FeSO_4_. All determinations were conducted in triplicate, and the mean values were finally expressed.

## 3. Results

### 3.1. Metabolome Profiling of C. sativa Leaves

The chemical composition of leaves from five distinct regions—Sichuan (SC-Z and SC-B), Xinjiang (XJ), Inner Mongolia (IM) and Shaanxi (SAX)—was initially analyzed using UHPLC-QTOF MS in both positive and negative ionization modes. To generate representative samples for each region, equal amounts of all batches from the same area were combined to form a region-specific sample for subsequent metabolome profiling (Figure 1). The leaves were divided into two main groups based on their chemical composition: Sichuan-Xinjiang, and Inner Mongolia-Shaanxi, with some overlap. Notably, the unique band produced by the active CBDAS sequence was absent in XJ and IM samples; while samples from SC-B, SC-Z and SAX exhibited relatively higher levels of DNA containing active CBDAS sequences (Appendix A). The observed variations in chemical composition among the leaves were more reflective of geographical differences rather than genetic distinctions. To delve deeper into the difference in content distribution, LC-MS/MS data of *C. sativa* leaves from various regions were concurrently analyzed based on FBMN.

### 3.2. SQFBMN-Based Cannabinoids and Flavonoids Characterization in Leaves

To investigate the chemical variations among the different regions, the semi-quantitative FBMN (SQFBMN) utilizing ionic strength with the clustered compounds was employed. Cannabinoids and flavonoids significantly influenced the pharmacological properties of *C. sativa*. Cannabinoids were the characteristic components of *C. sativa*, which were highly expressed in the plant inflorescence, but at relatively lower levels in the leaf and the stem. In terms of flavonoids, the total flavonoid content in the leaves exceeded that in other parts of the plant (inflorescence and stem bark) [26]. The relative contents of cannabinoids and flavonoids in leaves, analyzed using SQFBMN, were selectively extracted to facilitate rapid comparison of composition between different samples. The network of cannabinoids (Cluster A, Figure 2), flavonoid-*O* glycosides (Cluster B, Figure 2), and flavonoid-*C* glycosides (Cluster C & D, Figure 2) were found. Combined with the information on MS/MS and reference standards, the compounds in the network were further identified.

#### 3.2.1. Identification of Cannabinoids

In the process of identifying cannabinoids in *C. sativa* leaves, 15 compounds (A1–A15) were found in the network (Cluster A, Figure 2). A1, A4, A11 and A12 were identified as CBCA, Δ^9^-THCA A, CBDA and CBD by comparing them with reference standards. Firstly, the cleavage pathway of reference standards was analyzed in Appendix A, summarized, and then combined with FBMN clustering to complete the component annotation. A16 can be annotated as Δ⁹-THC in network and confirmed with the reference standard. However, it is not connected to any point, partly because of the extremely low abundance of Δ⁹-THC in *C. sativa* leaves from various regions. Since sufficient characteristic fragment ions were not generated during mass spectrometry (MS/MS) analysis, so it was not classified in the cluster of cannabinoids.

In the network—starting from A1 for analysis—A2, A6 and A7 are close to it. A2 and A1 differed by 16 Da (CH_4_), it was speculated that the substituent at the C-3 site of A2 is consistent with the pentyl (C_5_H_11_). The oxygen-containing six-membered ring of A2 is an open ring which is different compared with the A1 structure. It is preliminarily speculated that A2 is CBGAM, which is also consistent with the results reported in the literature [27]. Here, the distance between A6 and A1 was short, indicating the structure of A6 is similar to CBCA. The difference of [M − H]^−^ ions and fragments between A1 and A6 was 28 Da (CH_2_); thus A6 was inferred as CBCVA. Similarly, the [M − H]^−^ of A7 was 28 Da larger than that of A2, which was inferred as CBGA-C4 based on MS/MS fragment information. A13 was close to A6 in the network, and the [M − H]^−^ of A13 was 16 Da greater than that of A6. The quasi-molecular ion of A3 is 28 Da less than that of A4, as well as the fragment ions between A3 and A4 in MS/MS. Compared with A1/A4, the substituents at the C-3 site of A3 and A1/A4 were different. Therefore, A3 was identified as Δ^9^-THCV A, which was a propyl (C_3_H_7_) on 3-position instead of a pentyl (C_5_H_11_) on Δ^9^-THCV A. A14 and A3 have the same retention time, and by comparing the MS/MS spectrum with A3, A14 was identified as a fragment of Δ^9^-THCV A. A8 is both directly connect to A3, indicating that they belong to THCs. Comparing their MS/MS spectrum, A8 was identified as Δ^9^-THCAM A. A15 and A3 had shorter distances in the network, and the [M − H]^−^ of A15 was 28 Da (CH_2_) less than that of A3. Comparing their MS/MS, A15 was identified as THCA-C1. According to the references, A12 was inferred as CBC. Node A12, connecting to A4, was identified as CBD with the reference standard. A10 is connected to A11 and A12 in the network. Meanwhile, A10 and A11 had the same retention time; in which A10 was identified as the fragment of CBDA by comparing the MS/MS spectrum with A11. The mass–charge ratio of [M − H]^−^ and fragments of A9 was 28 Da smaller than that of A11, indicating a difference of CH_2_; so A9 was inferred as CBDVA.

#### 3.2.2. Identification of Flavonoid-O Glycosides

For annotation of flavonoid-*O* glycosides in *C. sativa* leaves, 11 compounds (B1–B11) were matched in the network database (Cluster B, Figure 2). Flavonoid glycoside is mainly composed of flavonoid aglycone and glycoside; the identification of which requires the determination of flavonoid aglycones according to the characteristic fragment. B3 and B4 were identified as apigenin-7-*O*-GluA and luteolin-7-*O*-GluA by comparison with the reference standards. In the positive ion mode, the MS/MS fragments of the compounds mainly included *m/z* 271, 285, 287 and 301. Combined with the MS/MS fragment information of B3 and B4, *m*/*z* 271 and 287 were preliminarily determined as quasi-molecular ions of apigenin and luteolin, respectively. The *m*/*z* 285 is 14 Da (CH_2_) more than *m*/*z* 271, and *m*/*z* 301 is 14 Da (CH_2_) plus *m/z* 287. According to the literature report [28], *m*/*z* 285 and 301 were the [M + H]^+^ ions of acacetin and chrysoeriol, individually.

Flavonoid-*O* glycosides were prone to neutral loss of glycosyl; the neutral loss ions of *m/z* 146 Da, 162 Da, and 176 Da were the characteristic fragments. With the help of fragment information of flavonoid aglycone, B5, B6 and B7 were monoglycosides; namely, Chrysoeriol-7-*O*-GluA, Acacetin-7-*O*-GluA and Chrysoeriol-7-*O*-GluA isomer. In addition, B1, B2, B9, B10 and B11 produced two glycosyl fragment ions; so they were inferred as flavonoid diglycosides, which were finally identified as Luteolin-*O*-GluA-Glc, Apigenin-*O*-GluA-Glc, Apigenin-*O*-GluA-Xyl, Luteolin-*O*-GluA-Xyl and Chrysoeriol-*O*-GluA-Glc isomer.

#### 3.2.3. Identification of Flavonoid-C Glycosides

In addition to flavonoid-*O* glycosides, we have also identified several flavonoid-C glycosides within the network (Cluster C & D, Figure 2). Cluster C is composed of four compounds (C1–C4). Similarly, taking C2 as an example, it was identified as vitexin by comparing it with the reference standard. Flavonoid-C glycosides had the same neutral loss in MS/MS, such as 30 Da (CH_2_O), 60 Da (C_2_H_4_O_2_), and 90 Da (C_3_H_6_O_3_), which were the characteristic fragments of flavonoid-*C* glycosides. Combining the mass–charge ratio of the node compounds, it was supposed that this cluster of compounds was flavonoid-*C* monoglycosides; then C1, C3 and C4 were identified as Chrysoeriol-*C*-Glc, Acacetin-*C*-Glc and Luteolin-*C*-Glc, respectively.

A total of seven compounds (D1–D7) were discovered in the network (Cluster D, Figure 2), among which D1 and D3 were identified as vitexin-4″-*O*-glucoside and vitexin-2″-*O*-rhamnoside by comparison of the reference standards; both of them were flavonoid glycosides. According to that, the other compounds in Cluster D were inferred flavonoid-C/O-diglycosides, which were identified by combining the fragments in MS/MS. D2, D4, D5, D6 and D7 were inferred as Luteolin-*C*-Glc-*O*-Glc, Acacetin-*C*-Glc-*O*-Glc, Luteolin-*C*-Glc-*O*-Rha, Chrysoeriol-*C*-Glc-*O*-Glc and Acacetin-*C*-Glc-*O*-Rha separately.

### 3.3. SQFBMN-Based Cannabinoids and Flavonoids Annotation in Leaves among Five Regions

The SQFBMN network of cannabinoids and flavonoids in leaves from five regions was established. Based on the network of SQFBMN (Figure 2), the chemical components in the leaves were mainly classified into cannabinoids, flavonoid-*O* glycosides and flavonoid-*C* glycosides.

#### 3.3.1. Cannabinoids in *C. sativa* Leaves from Five Regions

The proportion of different cannabinoids in *C. sativa* leaves varies depending on the region where the samples were collected. The content of A1 (CBCA) and A16 (Δ^9^-THC) in *C. sativa* leaves from different regions is similar. Besides, the content of A2 (CBGAM), A10 (CBDA fragment), A11 (CBDA), A12 (CBD) and A8 (Δ^9^-THCAM A) in Sichuan (SC-B and SC-Z) samples was relatively high. While the content of THCs, such as A3 (Δ^9^-THCVA), A5 (Δ^9^-THCA-C4), A14 (Δ^9^-THCVA fragment) and A15 (THCA-C1) was relatively high in SAX samples. In addition, the content of A6 (CBCVA) and A7 (CBGA-C4) in SAX samples was also higher. *C. sativa* from SAX, SC-Z and SC-B were both fiber-type, but the chemical composition distribution was opposite and complementary. Excluding the influence of genotype, the geographical distribution of cannabinoid components may be related to biogenic synthesis pathways, which were influenced by the growing environment. 

Samples from XJ and IM are both high expressions of the THCA gene, but their relative contents of Δ^9^-THCs are unidentical. The content of CBDs such as A9 (CBDVA), A10 (CBDA fragment), A11 (CBDA) and A12 (CBD) in *C. sativa* leaves of IM was very low. The content of A3 (Δ^9^-THCVA), A5 (Δ^9^-THCA-C4), A6 (CBCVA), A7 (CBGA-C4) and A14 (Δ^9^-THCVA fragment) in IM was significantly higher than that in XJ. It was obvious that the alkyl substitutions of cannabinoids in the C-3 position of resorcinol in the leaves from IM were mostly propyl and butyl, which were similar to SAX. This is mainly because the Inner Mongolia and Shaanxi regions are geographically close. 

#### 3.3.2. Flavonoid-O Glycosides in *C. sativa* Leaves from Five Regions

Eleven flavonoid-*O* glycosides (B1–B11) were compared in the SQFBMN network of *C. sativa* leaves from five places (Figure 2, Cluster B); the identification procedure is not detailed here but the results are displayed in Appendix A. It was obvious that the content of flavonoid-*O* glycosides in *C. sativa* leaves from IM was relatively high, followed by XJ and SAX. According to the distance of nodes, the flavonoid-*O* glycosides were mainly divided into two categories in the network; namely, flavonoid-*O* monoglycosides and flavonoid-*O* diglycosides. Flavonoid-*O* diglycosides, such as B1 (Luteolin-*O*-GluA-Glc), B2 (Apigenin-*O*-GluA-Glc), B8 (Apigenin-*O*-GluA-Rha), B9 (Apigenin-*O*-GluA-Xyl), B10 (Luteolin-*O*-GluA-Xyl) and B11 (Chrysoeriol-*O*-GluA-Glc isomer) were mainly distributed in IM. While flavonoid-*O* monoglycosides were evenly distributed in the three provinces mentioned, including B3 (Apigenin-7-*O*-gluA), B4 (Luteolin-7-*O*-gluA), B5 (Chrysoeriol-7-*O*-gluA) and B6 (Acacetin-7-*O*-gluA).

From the perspective of molecular networks, the types and contents of flavonoid-*O* glycosides were more distributed in the leaves of *C. sativa* from IM, followed by XJ and SAX. Flavonoid-*O* diglycosides were mainly distributed in IM, while flavonoid-*O* monoglycosides were distributed in three regions. These three provinces were located in the north of China, while Sichuan was located in the southwest of China, with a very low content of flavonoid-*O* glycosides; this was significantly different from the aforementioned three provinces.

#### 3.3.3. Flavonoid-C Glycosides in *C. sativa* Leaves from Five Regions

As shown in Figure 2, Cluster C was flavonoid-*C* monoglycosides, and Cluster D was flavonoid-*C*/*O*-diglycosides. Likewise, the retention time, molecular formula, identification results and other information on node compounds are shown in Appendix A. The content of flavonoid-*O* glycosides in Sichuan (SC-Z and SC-B) was much lower than that in other regions, while flavonoid-*C* glycosides in *C. sativa* leaves were mainly distributed in Sichuan. The content of flavonoid-C glycosides in the SC-Z and SC-B was basically the same, except for D6 (Chrysoeriol-*C*-Glc-*O*-Glc) and C1 (Chrysoeriol-*C*-Glc); both chrysoeriol glycosides were much higher in the leaves of SC-B than in SC-Z. In general, the content of flavonoid-*C* glycosides in *C. sativa* leaves of SC-B was slightly higher than that of SC-Z.

The geographical distribution trend of flavonoid glycosides in *C. sativa* leaves may correlate with the varying latitudes of the regions. Lower-latitude regions like Sichuan exhibit a prevalence of flavonoid-*C* glycosides, whereas in regions at higher latitudes—such as Xinjiang, Inner Mongolia and Shaanxi—they predominantly contain flavonoid-*O* glycosides. Based on the distinct glycoside bond atoms, the distribution of flavonoid-*C* glycosides was predominantly in Sichuan, while flavonoid-*O* glycosides were primarily found in the other three regions. Furthermore, in terms of the glycosyl type of flavonoid glycosides (glucose or glucuronic acid), the content of flavonoid glucoside is higher in Sichuan, while that of flavonoid glucuronide in Inner Mongolia is higher. In summary, the content of flavonoid glycosides in *C. sativa* leaves is influenced by geographical location.

### 3.4. Quantitative Analysis of Multiple Components in C. sativa Leaves

Due to significant differences in the relative content of flavonoids in *C. sativa* leaves from different regions, we expanded the sample size and sources to conduct an accurate quantitative analysis of the main flavonoids in *C. sativa* leaves. In addition, Δ^9^-THC and CBD were also considered key components for distinguishing the chemical types of *C. sativa*. Therefore, we further established a rapid and sensitive quantitative method for the simultaneous determination of multiple components in *C. sativa* leaves.

#### 3.4.1. Development and Validation of Quantitative Methods

Based on the relative content results of SQFBMN analysis of chemical components in *C. sativa* leaves, 11 flavonoids and two cannabinoids were selected as quantitative components. The structures of 13 analytes and internal standard compounds are shown in Figure 3. In reference to our prior investigation, the mobile phase consisting of 0.1% formic acid water and acetonitrile exhibiting excellent peak morphology and compatibility with the mass spectrometer was selected. To further reduce the analytic time, the mobile phase gradient was fine-tuned. Finally, the 13 targeted analytes with IS could be well separated within 20 min.

The multiple-reaction monitoring (MRM) mode was performed for quantification based on the optimized ion transitions of each analyte. To optimize the MRM conditions, ESI ion polarity, the parameters of Q1 voltage, Q3 voltage and collision energy were optimized to obtain the richest relative abundance of the parent and product ions. The optimized product ions and other MS parameters of all 13 analytes and IS are shown in Appendix A and the typical MRM spectrum of the 14 compounds is shown in Figure 4A.

The analytical method was validated for the linearity, limit of detection (LOD), limit of quantification (LOQ), precision, repeatability, stability and recovery of 13 compounds. All calibration curves show good linear regression (R^2^ > 0.991) within the tested ranges; the LODs (S/N = 3) and the LOQs (S/N = 10) for the 13 compounds are in the range of between 0.0021~0.7219 ng/mL and 0.0156~2.4612 ng/mL, showing a high sensitivity. The results of the linear calibration curve with R^2^, linear range, LOQ and LOD are shown in Appendix A. The RSD values of repeatability and stability analysis were not above 13.79%. The intra-day and inter-day precision was assessed within the same day and on 3 consecutive days, and the RSD value did not exceed 11.03%. Recovery analysis was performed by adding the 50% level of reference solution as the sample during the extraction process to evaluate the accuracy. The recoveries of the targeted compounds varied from 80% to 105% with RSD < 15.00%. The results of precision, repeatability, stability and accuracy are shown in Appendix A.

#### 3.4.2. Quantification of 13 Compounds in *C. sativa* Leaves from Different Regions

Using the established analysis method, the content of 73 batches of *C. sativa* leaves collected was determined. The specific results are shown in Appendix A. The content stacking diagram of different batches of *C. sativa* leaves is shown in Figure 4B. 

Among the 73 batches of *C. sativa* leaves samples, batches L02~L24 (GX, SC-B and SC-Z) predominantly contained Vitexin-4″-*O*-glucoside, while batches L25~L73 (QH, SAX, XJ, and IM) exhibited a significant presence of Apigenin-7-*O*-glucuronide and Luteolin-7-*O*-β-D-glucuronide. The former is flavonoid-*C* glycoside, while the latter two are flavonoid-*O* glucuronides, which is consistent with the rules summarized by FBMN. The total flavonoid content of 73 batches of *C. sativa* leaves samples ranged from 0.98 to 24.58 mg/g. The total flavonoid content of batch L66 (XJ) was the highest, and the content of batch L01 (YN) was the lowest. The flavonoid glucoside content ranged from 0.005 to 14.31 mg/g. The content of flavonoid glucuronide was lower than 23.37 mg/g, and the content of flavonoid glucoside Diosmetin 7-*O*-glucoside in batches L14~L25 was lower than the detection limit; Vitexin-4″-*O*-glucoside and Vitexin-2”-*O*-rhamnoside was not detected in batches L25~L44 and L57~L73; the content of Vitexin in batches L25~L26, L28~L44 and many batches from XJ was below the detection limit; the glycosides Apigenin-7-*O*-glucuronide and Luteolin-7-*O*-β-D-glucuronide were not detected in batches L01~L21. According to the content of each region, the content of flavonoid glucuronide: XJ > QH > SC (SC-B and SC-Z), showing obvious regional regularity, while the trend of flavonoid glucuronide is the opposite; this verifies the previous traceability research based on SQFBMN. Further analysis showed that YN, SC-B, SC-Z and GX (southwest regions) were mainly flavonoid glucuronides, while those in SAX, IM, QH and XJ (northwest regions) were mainly flavonoid glucuronides (Appendix A).

For the determination of cannabinoid content, the content of Δ^9^-THC ranges from 0.01 to 3.72 mg/g; the highest content is in batch L01 (YN) and the lowest is in batch L54 (SAX), with an average value of 0.32 mg/g for each batch. The content of CBD ranges from 0 to 7.73 mg/g, and is the highest in batch L17 (SC-Z) and is very low in L02, L03 (GX) and L72, L73 (IM) batches, which cannot be quantitatively analyzed. The average value of each batch is 0.74 mg/g. Overall, the content of cannabinoids (calculated by CBD and Δ^9^-THC) in *C. sativa* leaves from YN, SC-B and SC-Z was relatively high. From Appendix A, it was evident that the CBD content in *C. sativa* leaves from SC-B and SC-Z was significantly higher than that of Δ^9^-THC, while *C. sativa* leaves from IM, YN and GX had significantly higher Δ^9^-THC content than CBD.

### 3.5. Antioxidant Activity of C. sativa Leaves from Different Regions

Traditionally, the *C. sativa* plant has been renowned for its cannabinoids: a specific chemical class predominantly found in the female inflorescence. Of particular interest, both economically and medically, are the cannabinoid-rich female inflorescences; while the other plant parts such as the leaves hold no significance in medicinal production. Studies have indicated that *C. sativa* leaves are abundant in antioxidant compounds such as flavonoids and phenolics, and have demonstrated anti-inflammatory properties [29]. Given that the leaf is a pivotal part of the plant, conducting thorough research into its bioactivity and chemical composition could enhance the medicinal value of *C. sativa*.

To comprehensively characterize the antioxidant activity in vitro of 73 collected *C. sativa* leaves, DPPH, ABTS and FRAP assays were used simultaneously for the test. As shown in Figure 5 and Appendix A, most of the *C. sativa* leaves show good antioxidant activity. Though there is little difference among the three methods, the *C. sativa* leaves from SC, GX, QH and IM exhibited relatively high antioxidant activity, which are both more than 0.5 mmol/L in DPPH and ABTS assay, and more than 0.4 mmol/L from FRAP assay. Following are leaves from XA and XJ; the mean activities were from 0.22 to 0.47 mmol/L, respectively. The DPPH, ABTS and FRAP values of leaves from YN are 0.16, 0.16 and 0.066 mmol/L, respectively, indicating that these leaves exhibited the lowest antioxidant activity among all of the samples. The *C. sativa* from GX, QH and XJ are wild, and the others are *C. sativa* cultivated for industrial or medicinal purposes under legal supervision. Although accurate identification of the varieties of the *C. sativa* samples may not be possible, it is important to note that the antioxidant activity of the leaves is not related to the chemical type of samples, but rather to their components, which are impacted by the growth environment.

### 3.6. Correlation Analysis between the Total Content of Quantitative Compounds and Activity in C. sativa Leaves

Studies have shown a high correlation between the antioxidant activity of *C. sativa* and the contents of flavonoids [30]. To further investigate this, a bivariate correlation analysis was conducted between the total content of quantitative compounds in *C. sativa* leaves and their antioxidant activity, as shown in Appendix A. Overall, a significant positive correlation (*p* < 0.001) between the total flavonoids and cannabinoids and the antioxidant activity of the *C. sativa* leaves. However, Figure 6A reveals that while the sample L07 (SC-B) had the highest total content of flavonoids and cannabinoids, its antioxidant activity was not the most potent, suggesting that the factors influencing the antioxidant activity of *C. sativa* leaves are complex.

To delve deeper into which quantitative components are most critical for antioxidant activity, a partial least squares regression (PLS) was applied to identify the primary active antioxidant compounds. The PLS model, constructed from data on 73 batches from different regions, facilitated a correlation analysis between chromatographic data and antioxidant activity. Matrix X was compiled from the content of 13 compounds in these batches, while matrix Y represented the three activity indicators. As shown in Figure 6B, the importance of the X variable in the model can be summarized as the variable importance for the projection (VIP) value (general threshold > 1.0). In addition, to prevent overfitting of the model, we perform Permutation validation on the model. The criterion for judging whether the model has not been fitted is that the intercept of R^2^ on the Y-axis is less than 0.4, and the intercept of Q^2^ on the Y-axis is less than 0.05. The displacement test results (Figure 6C) show that all Q^2^ and R^2^ are lower than the original point on the right, and the Q2 regression intercept is less than 0.05, indicating that the model has not overfitted. 

The correlation coefficients listed in Appendix A indicate the strength of the relationship between each component and its activity. Notably, Vitexin-4″-*O*-glucoside and Luteolin-7-*O*-glucuronide exhibited high VIP values and strong correlation coefficients with sample biological activity, highlighting their substantial impact on antioxidant effectiveness. Additionally, the cannabinoid compound CBD showed a VIP value of 0.747, underscoring its significant contribution and robust antioxidant activity.

## 4. Discussion

*C. sativa* leaves are rich in cannabinoids and flavonoids, which exhibit significant antioxidant properties. To reveal the antioxidant effects and accurately identify the active ingredients that play a key role, establishing a profiling and notation method for *C. sativa* leaves from different regions is imperative. Their different components and distribution tendencies were clarified with the help of UHPLC-Q-TOF and the SQFBMN. In total, 16 cannabinoids and 57 flavonoids in *C. sativa* leaves were rapidly and systematically characterized based on FBMN and HRMS. Unlike FBMN and MN, SQFBMN can not only increase the exposure of components with large content differences in the sample but also visually and quickly compare the content differences between samples. The distribution of cannabinoids and flavonoids in *C. sativa* leaves from different regions indicated apparent geographic tendency.

From the molecular network results, the cannabinoids in leaves from SC-B, SC-Z and SAX had different preferences for alkyl substitution at the C-3 position. C5 alkyl side chain accumulation was prevalent in leaves from Sichuan, while Shaanxi mainly produced cannabinoids with a C-3 propyl (C_3_H_7_) or a C-3 butyl (C_4_H_9_) alkyl side chain. This phenomenon may be related to the biosynthetic pathway of cannabinoids, which is mainly divided into two sub-pathways. The methylerythritol phosphate pathway leads to the production of the isoprenoid intermediate geranyl diphosphate, determining the type of cannabinoids. While the alkylresorcinolic acid pathway leads to the production of the cannabinoid intermediate olivetolic acid, affecting the type of C-3 substituents in cannabinoids. [31] The results of the subsequent multi-component quantitation further indicated that flavonoid glycosides were mainly distributed in low-latitude regions such as Sichuan, Yunnan and Guangxi; while flavonoid glucuronides were mainly distributed in high-latitude regions such as Shaanxi, Inner Mongolia, Qinghai and Xinjiang.

Our previous research based on SQFBMN reveals that the flavonoids distributed in leaves could be a potential chemical marker of *C. sativa*. Flavonoids are a large group of plant secondary metabolites which almost certainly present in all terrestrial plant species. They play important physiological and biochemical functions in plant life such as the protection of plant tissues against the harmful effects of ultraviolet radiation. Meanwhile, they exhibit diverse pharmacological effects such as antioxidant and anti-inflammatory effects, which are beneficial to human health. Therefore, to reveal the correlation between chemical components and antioxidant activity of *C. sativa* leaves from different regions, more samples were collected for multi-component quantitative analysis and antioxidant activity testing. A quantitative method based on UHPLC-QQQ MS was established for the quantification of 11 main flavonoids and two cannabinoids in 73 batches of *C. sativa* leaf samples. Three methods were used for antioxidant activity testing, followed by correlation analysis between component content and antioxidant activity. The results indicated that Vitexin-4″-*O*-glucoside, luteolin-7-*O*-glucoside and CBD have high contribution and good antioxidant activity.

## 5. Conclusions

In this study, a total of 16 cannabinoids and 57 flavonoids were annotated from *C. sativa* leaves based on the untargeted metabolite profiling method combined with a semi-quantitative FBMN (SQFBMN) approach, which also revealed the chemical diversity in *C. sativa* leaves among various regions. Then the correlation between the chemical composition and the antioxidant activity of *C. sativa* leaves was investigated. The antioxidant activity exhibited by *C. sativa* leaves is correlated with the content of cannabinoids and flavonoids inside. In summary, in-depth analysis of chemical components is indispensable in the field of research on the pharmacological activity of phytomedicines. At the same time, further exploration of the impact of growth environmental factors on secondary metabolites is of crucial significance for achieving comprehensive, efficient, and rational utilization of their medicinal resources.

## Figures and Tables

**Figure 1 antioxidants-13-00749-f001:**
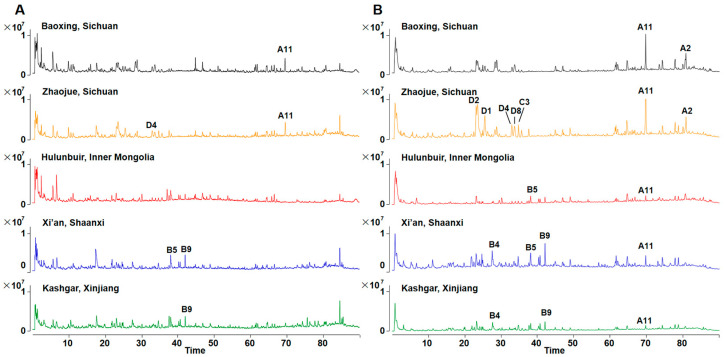
The total ion chromatograms (TIC) of leaves of Cannabis sativa from five places acquired by UHPLC-Q-TOF-MS. (**A**) Positive mode; (**B**) negative mode.

**Figure 2 antioxidants-13-00749-f002:**
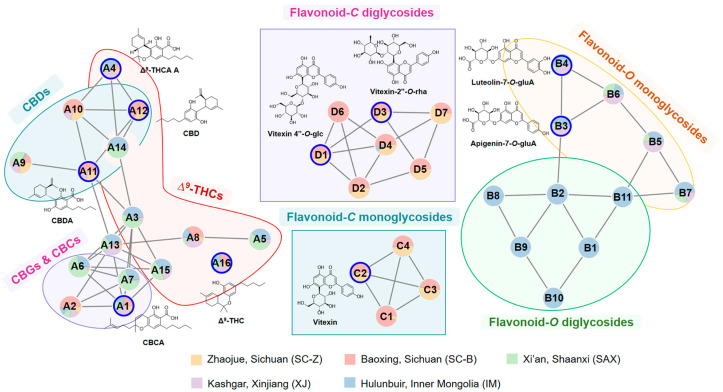
The selected molecular network of cannabinoids and flavonoid glycosides in *C. sativa* leaves from five regions.

**Figure 3 antioxidants-13-00749-f003:**
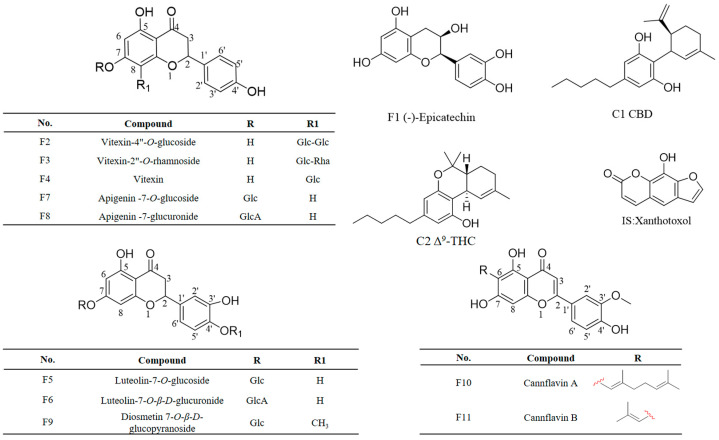
The structures of 13 analytes and IS.

**Figure 4 antioxidants-13-00749-f004:**
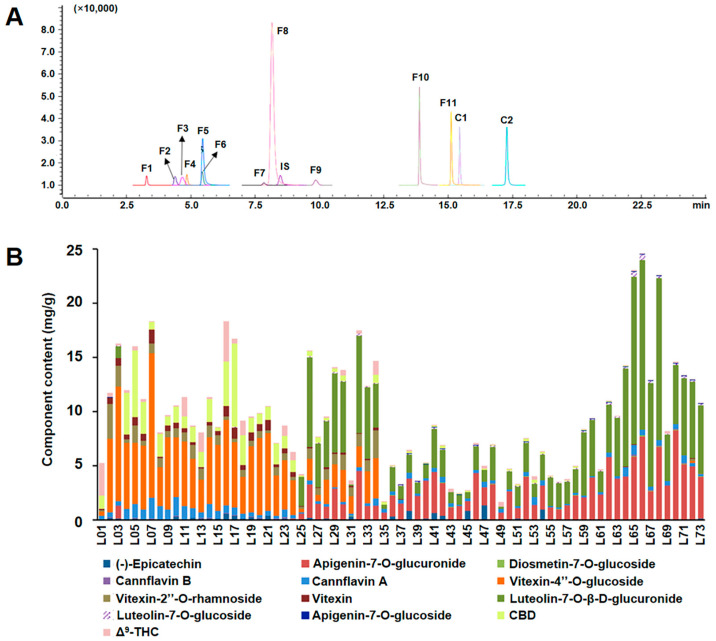
Establishment of quantitative methods and determination of 73 batches of samples. (**A**) Typical MRM chromatograms of the 13 analytes and IS determined in *C. sativa* leaves; (**B**) Quantitative results of 13 chemical components in 73 batches of *C. sativa* leaves.

**Figure 5 antioxidants-13-00749-f005:**
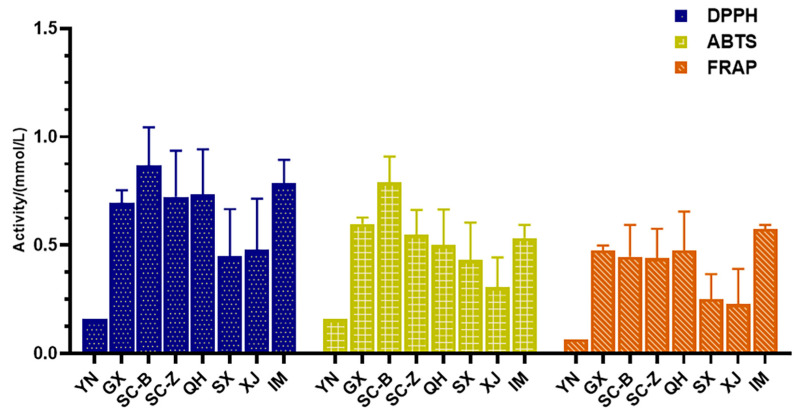
Antioxidant activity of *C. sativa* leaves from different regions. (YN: Yunnan sample; SC-B: Sichuan Baoxing sample; SC-Z: Sichuan Zhaojue sample; GX: Guangxi Nanning sample; XA: Shaanxi Xi’an sample; IM: Inner Mongolia Hohhot sample; QH: Qinghai Xining sample; XJ: Xinjiang Kashgar sample).

**Figure 6 antioxidants-13-00749-f006:**
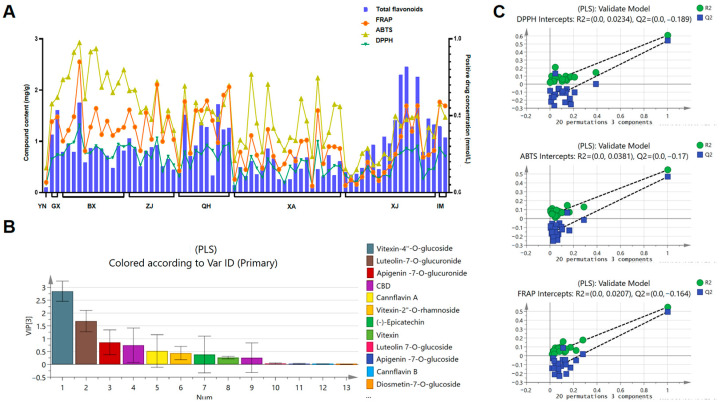
Analysis results of correlation between antioxidant activity and compound content. (**A**) Total flavonoid content and antioxidant activity; (**B**) VIP values for PLS analysis of the main flavonoids and cannabinoids compounds and their antioxidant activity; (**C**) Results of permutation test on PLS model. (YN: Yunnan sample; BX: Sichuan Baoxing sample; ZJ: Sichuan Zhaojue sample; GX: Guangxi Nanning sample; XA: Shaanxi Xi’an sample; IM: Inner Mongolia Hohhot sample; QH: Qinghai Xining sample; XJ: Xinjiang Kashgar sample).

## Data Availability

The data are available in the Appendix A.

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
