# Peer review of "Feature-Based Molecular Network-Assisted Cannabinoid and Flavonoid Profiling of Cannabis sativa Leaves and Their Antioxidant Properties"

_antioxidants, 2024, doi:10.3390/antiox13060749_

Round 1

Reviewer 1 Report

The manuscript entitled “Feature-based molecular networks-assisted cannabinoids and flavonoids profiling of Cannabis sativa leaves and their antioxidant properties” by Chen et al describes the untargeted phytochemical analysis and comparison of different C. sativa leaves extracts with different geographic origin. Moreover the authors reported the quantification of the main components and the antioxidant bioactivity evaluation of the extracts. The results are interesting and the methods are generally appropriate, but the text needs certain improvement before it is accepted for publication in the Journal. The degree of novelty of these results can be considered appropriate for the Antioxidants readers.

The main comment concerns the C. sativa samples. The relative content of phytocannabinoids in cannabis is highly variable and based on the relative abundance of THCA, CBDA and CBGA, five cannabis plant chemotypes (I-V) can be identified. The authors did not mention this. Which chemotype did they analyse? Do the quantitative analyses performed and the cannabinoid quantification results correspond to the analysed chemotype?  Please comment on this.

The structure of the CBD in Figures 2 and 3 needs to be corrected.

Is the molecular network shown in Figure 2 the complete network or have the authors only reported selected networks? This should be specified in the text and in the figure caption. Also, could the authors provide the link to the molecular network on the GNPs website to access and visualise it?

Overall, the English in this manuscript is rough and the language issues need to be addressed at the review stage.

Reviewer 2 Report

The subject of this study is interesting and combines the use of an algoruthm with the different compositions of C. Staiva. This may be useful to know and predict plant composition according to its location.

The work is well stuctured, but I believe that the authors should reinforce why it is important tocross this data with location, what are the benefits of using this combined approach and how it can help beyond the scope of this work. IS it to predict the antioxidant potential of a certain harvest? Is it to determine which location suits better to a given application? This should be well patent in the text.

Abstract- the abbreviations FBMN (SQFBMN) should be preceeded by their meaning

Line 49, rephrase this sentence, it should end as applications or products or something similar

Line120, are the S/L ratios correct? Also extraction times are 1 and 2 hours? minutes? this should be rephrased.

Line 212- the meaning of the abbreviations DPPH, ABTS and FRAP should be before them 

LIne 550, what is "good antioxidant activity"? Is it a comparison? A meausure? This should be rephrased

Round 2

Reviewer 1 Report

The present version of the manuscript has been revised properly based on the previous comments.

The present version of the manuscript has been revised properly based on the previous comments.